# Heavy Metal Transporters-Associated Proteins in *Solanum tuberosum*: Genome-Wide Identification, Comprehensive Gene Feature, Evolution and Expression Analysis

**DOI:** 10.3390/genes11111269

**Published:** 2020-10-28

**Authors:** Guandi He, Lijun Qin, Weijun Tian, Lulu Meng, Tengbing He, Degang Zhao

**Affiliations:** 1The Key Laboratory of Plant Resources Conservation and Germplasm Innovation in Mountainous Region (Ministry of Education), Institute of Agro-Bioengineering and College of Life Sciences, Guizhou University, Guiyang 550025, China; gs.gthe15@gzu.edu.cn (G.H.); ljqin@gzu.edu.cn (L.Q.); 2Agricultural College, Guizhou University, Guiyang 550025, China; gs.tianwj18@gzu.edu.cn (W.T.); gs.llmeng18@gzu.edu.cn (L.M.); 3Institute of New Rural Development of Guizhou University, Guiyang 550025, China; 4Guizhou Academy of Agricultural Science, Guiyang 550025, China

**Keywords:** gene duplication, phylogenetic analysis, abiotic stress, expression analysis

## Abstract

Plants have evolved a number of defense and adaptation responses to protect themselves against challenging environmental stresses. Genes containing a heavy metal associated (HMA) domain are required for the spatiotemporal transportation of metal ions that bind with various enzymes and co-factors within the cell. To uncover the underlying mechanisms mediated by *StHMA* genes, we identified 36 gene members in the StHMA family and divided them into six subfamilies by phylogenetic analysis. The *StHMA*s had high collinearity and were segmentally duplicated. Structurally, most StHMAs had one HMA domain, StHIPPc and StRNA1 subfamilies had two, and 13 StHMAs may be genetically variable. The StHMA gene structures and motifs varied considerably among the various classifications, this suggests the StHMA family is diverse in genetic functions. The promoter analysis showed that the *StHMA*s had six main cis-acting elements with abiotic stress. An expression pattern analysis revealed that the *StHMA*s were expressed tissue specifically, and a variety of abiotic stresses may induce the expression of StHMA family genes. The HMA transporter family may be regulated and expressed by a series of complex signal networks under abiotic stress. The results of this study may help to establish a theoretical foundation for further research investigating the functions of HMA genes in *Solanum tuberosum* to elucidate their regulatory role in the mechanism governing the response of plants to abiotic stress.

## 1. Introduction

Potato (*Solanum tuberosum*) is an important food crop worldwide and a good source of vitamins, minerals and dietary fiber as well as phytochemicals, which benefits human body as nutrients supplementary and antioxidants [1]. Due to its special geological conditions, Guizhou has most of its soil developed from carbonatite. The parent material that determines the Cd content in local soil remains quite high [2], which severely undermines the growth, yield, quality, and production of *S. tuberosum*.

During their growth and development, plants absorb such essential microelements as cuprum (Cu), zinc (Zn), and manganese (Mn), as well as some non-essential metal elements like Cadmium (Cd), plumbum (Pb), and mercury (Hg) [3]. When excessively accumulated, non-essential elements can not only affect the growth and development of plants, but also gather in the human body and undermine human health through the effect of food chain [4,5]. Recent studies have revealed that, during plants’ growth and development, a mechanism that can finely regulate extracellular metallic ion concentration is formed to guarantee the uptake of essential trace metal elements and reduce plants’ tolerance of non-essential heavy metal ions at the same time [6]. According to related studies, Cd absorption within plants involves a string of procedures, including Cd absorption by root, Cd loading/unloading by xylem, transport of Cd from xylem to phloem, Cd redistribution among aerial parts, such as stem and leaves, and Cd accumulation in seeds [7,8]. The transport of Cd in scattered vascular bundles at stem nodes from the xylem to phloem is considered as a key factor determining the Cd content in aerial parts [9]. In recent years, several genes or quantitative trait locus (QTLs) have been identified as being related to Cd absorption and transport, including heavy metal ATPase (HMA) [10], ATP-binding cassette (ABC) [11], iron transporter (IRT) [12,13], metal tolerance protein (MTP) [14], natural resistance associated macrophage protein (NRAMP) [15], cation-efflux transporter (CET) [16], cation/H+ antiporters (CAX) [17,18], Zn-regulated transporter-like protein (ZNT/ZIP) [19,20], and low-affinity cation transporter (LCT) [21]. Among those gene families, P-type ATPase is kind of ion pump that utilizes the energy resulting from ATP hydrolysis to achieve transmembrane transport of ions and is widely seen in bacteria, plants and human body [22]. It can be further divided into five subfamilies by the transported substrate, namely heavy metal ATPase (HMA/P1B), Ca^2+^-ATPase (P2A/P2B), H^+^-ATPase (P3A), ALA (P4), and P5-ATPase (P5) with unknown substrate specificity [23]. Among them, HMAs/P1B are some channel proteins that associate with plants’ absorption of trace metal nutrients, which function in xylem load in case of long-distance transport of metal ions by binding with metal ions in large quantities depending on P-type ATPase activity and completing the transport to aerial parts [24]. This family is able to achieve transport through cytomembrane by hydrolyzing ATP [22]. As revealed by studies concerning genetic functions of the HMAs family, for rice, *OsHMA2* located in plasma membrane is involved in the loading of Cd and Zn of xylem [25,26,27], *OsHMA5* takes a part in loading Cu onto the xylem of root and other organs [28], *OsHMA3* on tonoplast is charge of fixing Cd in vacuoles of the root [29], and *OsHMA4* acts to separate Cu into root vacuoles and limit its accumulation in seeds [30].

Different stresses such as cold, drought, and high salinity share some common features with regards to their impacts on plants and the ways by which plants perceive them. For instance, The stress sensing mechanisms, including receptor-like kinases, cytoskeleton-related mechanosensors, stretch-dependent ion (calcium) channels and redox-mediated system, act at multidimensional levels in order to integrate the perception of a specific environmental cue [31]. These signal transduction pathways lead to the activation of well characterized proteins involved in the biosynthesis of proteases, transporters, and chaperones as well as ROS detoxification enzymes (alternative oxidase, glutathione peroxidase, glutathione reductase, copper-zinc superoxide dismutase and glutathione S transferase) [32]. The molecular mechanisms and signal transduction pathways of the HMA gene family under abiotic and biotic stress remain poorly understand. The HIPP gene family containing HMA domain has made some progress against abiotic and biotic stress. Roles of HIPP proteins in heavy metal homeostasis and/or detoxification [33], cold- and salt-stress responses [34], virus long-distance movement [35], and plant–pathogen interactions [36] have been demonstrated. The plant perception of overall stress induces a coordinated response mediated by complex signaling networks [31]. Plants have evolved a finely tuned response system enabling coordinated cross-talk between specific defense pathways involving shared elements and common outputs [32]. However, there is a dearth of research in the HMA gene family.

This study sets out to analyze the genetic structure, chromosome distribution and phylogenetic relationship of HMA gene family in the hope to support the understanding of the evolution and diversification of HMAs in *S. tuberosum* species at a genome-wide scale. The potential functions of the potato HMA genes were predicted through their expression profiles of 6 abiotic stresses analysis, will lay a theoretical foundation for further study on the functions and abiotic stress (like heavy metal, high salt, high temperature, low temperature, flooding, and drought stress) resistant response mechanism of HMA gene family in *S. tuberosum*.

## 2. Results

### 2.1. Genome-Wide Identification, Chromosome Assignment, and Collinearity Analysis

In this study, 36 candidate *S. tuberosum* HMA genes, respectively, were identified by a hidden Markov model for the HMA domain. In keeping with their locations on the chromosome, these genes were named *StHMA1*-*StHMA36* in turn. A bioinformatic analysis of the amino acid sequences of these 36 family members showed that their molecular weights ranged from 7.69 to 138.88 kDa, and 19 genes were located on the minus strand with 70–1002 amino acids (StHMA6 remains the smallest with 70 amino acids, whereas StHMA9 has over 1000 encoded amino acids). The isoelectric points of HMA proteins are in the range of 4.72–9.98. The StHMA family members consist of 22 basic proteins, 13 acidic proteins and 1 neutral protein. The difference in isoelectric points indicates the diversity of the three-dimensional spatial structure of the protein sequence. Appendix A lists the physicochemical properties of the StHMA proteins. The forecast of subcellular localization suggests that in the HMA gene family of S. tuberosum, 26 StHMAs exist in the cytoplasm, 22 in the cell nucleus, 26 in the chloroplast, seven in the plasma membrane, 10 in the mitochondria, and nine in the cytoplasmic matrix; furthermore, StHMA30 and StHMA23 are in the vacuoles, StHMA9 is in the endoplasmic reticulum, StHMA1 and StHMA4 are in the peroxisome, and StHMA4 and StHMA9 are in the plasma membrane (Appendix A).

For *HMA* genes obtained from *S. tuberosum*, their locations on chromosomes were determined by searching the *S. tuberosum* database (http://plants.ensembl.org/Solanum_tuberosum/Info/Index). After all the sequences were located on *S. tuberosum*, a location map of those genes was constructed using Mapinspect software (Figure 1). An analysis revealed that 36 genes in this family are distributed on 11 chromosomes of *S. tuberosum*, with 1–8 being on each chromosome at varying locations. All the genes are primarily located at the two ends of the chromosomes. There are eight genes on chromosome 1, four genes on chromosomes 3, 4, 5 and 11, three genes on chromosomes 2 and 10, two genes on chromosome 6, and one gene on chromosomes 8, 9, and 12, respectively. However, no member of this gene family has been detected on chromosome 7. In summary, *StHMA*s are primarily located on Chr1-5 and Chr11.

Six pairs of genes exhibited massive chromosome multiplication events (segmental duplication) in keeping with *S. tuberosum* intraspecific collinearity analysis, and the pairs are *StHMA11* & *StHMA12*, *StHMA11* & *StHMA16*, *StHMA31* & *StHMA28*, *StHMA31* & *StHMA26*, *StHMA18* & *StHMA19*, and *StHMA18* & *StHMA22*. Tandem duplication is observed with *StHMA19* and *StHMA20*. The results indicate that some genes may cause HMA family members to multiply on different chromosomes through gene duplication, and repeats are localized to chromosomes 2–6, 9, and 10 (Figure 2). To further analyse the homologous relationships between *S. tuberosum* and other species in terms of the HMA gene family, six species were chosen for collinearity analysis, including *Solanum tuberosum*, *Arabidopsis thaliana*, *Ipomoea triloba*, *Nicotiana attenuata*, *Zea mays*, and *Oryza sativa* (Figure 3, Appendix A). Among 49 pairs of orthologous genes discovered in the analysis, *S. tuberosum* shares only one HMA orthologous gene with *O. sativa*, four with *Z. mays*, six with *N. attenuata*, 20 with *I. triloba*, and 18 with *A. thaliana*. Thus, it can be assumed that the collinearity between *S. tuberosum* and *A. thaliana* and *I. triloba* is more significant than that observed between *S. tuberosum* and *O. sativa*, or *Z. mays* and *N. attenuata*. In addition, HMA genes in both *I. triloba* and *A. thaliana* have corresponding paralogous genes in *S. tuberosum*, and they largely have more than two such paralogous genes in *S. tuberosum*. This association may have played a central role in the gene duplication of the HMA gene family. The foregoing results show that numerous *HMA*s could be produced by gene replication. We calculated the Ka/Ks ratios of the HMA homologous gene pairs to determine the selective pressure on the HMA gene family (Appendix A). Most of the gene pairs had Ka/Ks < 1. However, some homologous gene pairs showed Ka/Ks > 1, indicating that these genes might have been subjected to positive selective pressure.

### 2.2. Multiple Sequence Alignment, Phylogenetic Analysis, and Classification

To extend the study on the phylogenetic relationship among *HMA*s, 8, 12 and 22 HMAs were selected from *A. thaliana*, *O. sativa*, and *Solanum lycopersicum*, respectively, to be incorporated into a maximum likelihood phylogenetic tree together with the 36 HMA amino acid sequences identified from *S. tuberosum*. According to the structures and functions of these genes, they were divided into 6 subfamilies named StHIPPa, StHIPPb, StHIPPc, StCCH, StPAA1 and StRAN1. Some members that had been previously confirmed as RAN1, PAA1, CCH, and HIPP were assigned to corresponding subfamilies, including AtHMA7, AtHMA4, and AtHMA8 of *A. thaliana*, OsHMA2, OsHMA8, and OsHMA12 of *O. sativa*, and SlHMA3, SlHMA2, and SlHMA8 of *S. lycopersicum* (Figure 4).

When the amino acid sequences of different StHMA subfamilies were aligned (Figure 5), StHIPPc and StRNA1 subfamilies were determined to contain two highly conserved core sequences, “CXXC” (in which X indicates different amino acids) and metal binding domains (MBD), while StCCH, StHIPPa, and StHIPPb were observed to have only one. In contrast to other subfamilies, StRNA1 exhibits an A-domain “TGES” aa motif and a P-domain “DKTGT” aa motif at the C-end [37]. The A-domain regulates the absorption and release of heavy metal ions, similar to a gateway. In RAN1-1 of *A. thaliana*, when the A-domain exhibits variation, the transport power is notably impaired [38]. The amino acid sequence of the P-domain is homologous to the Haloacid dehalogenase (HAD) family, as they can both be phosphorylated [39,40]. Mutations of the sequence motif aspartic acid significantly inhibits the transport capacity of AtHMA3 [41]. Therefore, StHMA9 may exhibit a complete set of transport and detoxication functions against heavy metals. StPAA1 and StCCH display a more conserved structural domain. In the StHIPPa, StHIPPb, and StHIPPc families, MBD is usually characterized by mdCd/eC, iHCdgC and m/l/fHCegC aa, but variation may occur in 13 family members, including StHMA31 (with one MBD missing), StHMA27 (with one more cysteine residue; presented as CCSGC) and StHMA29 (with HCEGCaa changed to be HCPKCaa) (Figure 5). For more details, please see Figure 5. In the StHIPPb family, except for StHMA35 and StHMA30, the remaining six subfamilies are all located at the N-terminus. Taken together, these findings indicated that each subfamily has its own unique conserved MBD and may exert different functions.

### 2.3. Analysis of Conserved Motifs and Gene Structure

By referring to the HMA genetic structures and sequence motifs of *A. thaliana*, *O. sativa* and *S. lycopersicum*, the differences in the structures and sequence motifs of different HMA subfamilies of *S. tuberosum* were determined. The number of exons of the HMA gene family in *S. tuberosum* were determined to range from 1 to 9, which is in keeping with the classification of HMAs in each subgroup. Specifically, StRAN1 was determined to have the most exons followed by StHIPPc, StPAA1, StHIPPa, StHIPPb, and StCCH. StHMA9 was determined to have the highest number of exons. The members in the same branch exhibited a common property: exon length was determined to correspond with the number (Figure 6).

The motif distribution of HMA gene family members of *S. tuberosum*, *A. thaliana*, *S. lycopersicum*, and *O. sativa* was studied using MEME online software, and a total of 15 motifs were identified (Figure 6). As revealed by the results, the subclasses vary significantly. Motifs 1 and 3 were identified as HMA structural domains that exist in all StHMA genes. The amino acid sequence of StRAN1 is clearly higher than that of other subclasses, and it was also determined to have the highest number of motifs, including 10 specific motifs (Motifs 2, 4, 5, 7, 9, 10, 11, 12, 14, and 15). These motifs may function to regulate the absorption and release of heavy metal ions, undergo phosphorylation, and provide energy for transmembrane transport [37,42,43]. Thus, the StHMA proteins in StRAN1 may be completely functional in undertaking heavy metal transport and detoxication.

### 2.4. Expression Profiles in Various Tissues and Organs

To investigate the expression profiles of different genes at various transcriptomic levels, the expression profiles of HMA gene family RNA-seq data downloaded from the Potato Genome Sequencing Consortium 2001 were analysed. These data cover 36 genes from the HMA family expressed in 15 tissues, such as root, stem, leaf, flower, stamen and stolon, accompanied by a heat map (Figure 7). All HMA genes were determined to be expressed in at least one tissue or organ, while some were observed to be more strongly expressed in some specific tissues and organs. For instance, *StHMA 1/2/5/11/17/18/24/28/35* were observed to be expressed in roots, *StHMA 3/12/15/33/36* in stems, *StHMA 30/32* in leaves, *StHMA 2/4/8/9* in petioles, *StHMA 15/29* in ripe tubers, *StHMA 3/34* specifically in tuber sprouts, *StHMA25* in piths, pellicles and cortex of tubers, *StHMA 4/11/10* notably in tuber buds, *StHMA 6/13/17/21/26* notably in stolons, *StHMA 7/22* specifically in stem tips, and *StHMA 28/31* specifically in stamens. Even the same genes were observed to be expressed differently in various tissues and organs. For example, *StHMA8* and *StHMA18/27* were observed to be more highly expressed in the petiole and root, respectively, but less strongly expressed in other tissues and organs. Overall, the genes in the HMA family were determined to be more highly expressed in underground parts than in aerial parts. These genes were observed to be strongly expressed in roots, stolons, and petioles; these genes were observed to be least strongly expressed in flowers; and they were observed to be expressed similarly in the pith, surface, and cortex of tubers.

### 2.5. Analysis of Abiotic Stress-Related Cis-Acting Elements and Expression Pattern of Partial StHMA Members

A region 2000 bp upstream of the StHMAs’ genomic sequences was selected for an analysis of cis-elements in the abiotic stress-related promoter sequences of *StHMA*s (Figure 8). *StHMA*s have several cis-elements that are related to biotic stress. Among these elements, six cis-acting elements are predominant. Drought (MYB, MYC and MBS), low temperature (LTR), abscisic acid (ABRE), and disease resistance (W-box) were induced to cause stress reactions in plants. It was assumed that the HMA gene family of *S. tuberosum* would react to multiple abiotic stresses. As a result, 15 *StHMA* genes were chosen to be processed with 6 abiotic stresses. An analysis of qRT-PCR results indicates (Figure 9) that when processed with different abiotic stresses, *StHMA*s were distinctly expressed in various tissues. Compared with the control group, the expression levels of *StHMA16* and *StHMA31* stressed by CdCl_2_, that of *StHMA19* and *StHMA36* stressed by PEG6000, that of *StHMA17*, *StHMA31* and *StHMA36* stressed by flood, that of *StHMA 1/16/19/31* stressed by NaCl, that of *StHMA 2/4/19/31* stressed by 40 °C, and that of *StHMA 2/4/17/19/22/31/36* stressed by 4 °C displayed significant variations at one time point. When stressed by specific abiotic factors, some genes exhibited higher transcription levels in specific tissues. For instance, in response to CdCl_2_ stress, *StHMA1* and *StHMA16* maintained high expression levels in aerial parts; in response to PEG6000 stress, *StHMA36* was highly expressed in roots; in response to 40 °C stress, *StHMA2* and *StHMA4* were highly expressed in stems and roots, respectively; in response to 4 °C stress, *StHMA19* was highly expressed in aerial parts, whereas *StHMA 8/17/25/36* were highly expressed in stems. Aside from those results, in all tissues, *StHMA31* expression was significantly elevated at all time points when stressed by Cu^2+^, Na^+^, 40 °C and 4 °C in contrast to the tissues in the control group. This gene may have been activated by several promoter elements to play roles in more than one abiotic stress. When processed with Cd^2+^, *StHMA 1/4/16/19/22/31/33/34/36* were observed to actively react to this stress at least one time point in different tissues (Figure 9). When processed with PEG6000, except for *StHMA19* and *StHMA36*, which present an active response to stress, all the remaining *StHMA*s were observed to remain actionless. When subjected to flooding, *StHMA 17/19/25/31/36* presented an active response to stress by being mostly expressed in the stem and root. In contrast, *StHMA 1/2/4/8/16/22/35* were not observed to be significantly elevated, regardless of time and location. When processed with Na^+^, *StHMA 1/2/4/8/16/19/22/31/34* in the root and stem were observed to actively respond to stress, *StHMA 16/19/22/31* in the leaf were determined to be highly expressed, and the expression levels of *StHMA 16/19/31* were observed multiplied by over 30-fold compared with those at 0 h. When stressed at 40 °C, *StHMA2*, *StHMA19,* and *StHMA31* displayed significantly high expression, and *StHMA2* was determined to be primarily expressed at 6 h. When stressed at 4 °C, except for *StHMA 2/4/16/22*, all other genes were observed to be remarkably expressed in the stem, and *StHMA19* and *StHMA31* were also determined to be highly expressed in the leaf. In summary, since some *StHMA* members were observed to respond to different abiotic stresses in varying degrees, it can be speculated that the *StHMA*s not only exerts certain effects on heavy metal stress, but also plays an important role in reacting to external abiotic stresses. Furthermore, the members of this family vary from one another in terms of their expression profiles and functional patterns.

## 3. Discussion

The HMA protein contains a heavy metal binding domain [37]. We first identified 36 StHMA genes from the HMA gene family of *S. tuberosum* and divided them into six classes according to their structural features (Figure 4).

A collinearity analysis of the *StHMA* family revealed 6 pairs of segmental duplication regions, including *StHMA11* & *StHMA12* and *StHMA11* & *StHMA16*, and tandem duplications, such as *StHMA10* & *StHMA20*. These gene duplications suggest that segmental and tandem sequences matter greatly in extending the species specificity of gene families [44]. This finding provides a new perspective for further functional genomic studies and molecular design-based breeding of *S. tuberosum*. Intraspecies collinearity analysis revealed 49 pairs of homologous genes; among them, 18 were in *A. thaliana*, six were in *N. attenuata*, four were in *Zea mays*, and one was in *O. sativa*, indicating different evolutionary branches between dicotyledons and monocotyledons during their phyletic evolution [45].

According to an alignment of HMA domain sequences, there is one core sequence in the HMA structural domain of each group, namely, the highly conserved CXXC (Figure 5). The StHMA subfamilies have their own unique conserved sequence motifs. For instance, StHMA9 has an A-domain “TEGS” aa motif and a P-domain “DKTGT” aa motif at the C-terminus that can regulate the absorption and release of heavy metal ions and promote phosphorylation [37]. These motifs indicate that StHMA9 may be functionally complete in terms of heavy metal transport and detoxication. During evolution, 13 members of the StHMA family, including StHMA31, StHMA27, and StHMA29, may undergo natural variation due to environmental stresses or genetic recombination and show some unique functions in StHMAs, which may be employed in future population genetics or functional genomics studies [46,47,48]. The genetic structure and sequence motif analysis demonstrated that genes in the same subfamily are relatively conserved (Figure 6), but StHMA family proteins vary significantly under different classifications. This finding may be attributed to the high number of exons in StHMAs. Exon shuffling and alternative splicing enrich the diversity of proteins in the StHMA family [49].

The findings of *StHMA*s expression profile analysis suggest that the expression levels of *StHMA*s are tissue-specific. This finding is in keeping with the results of previous studies [50,51]. Thus, these *HMA*s may be specifically involved in stress reaction-associated biological functions. *StHMA31* (*OsHIPP3-1*; *AtHIPP5*) has a higher transcription level in the stamen than other genes, while *StHMA11* (*AtPAA1*) is more heavily transcribed in tuber sprouts than other *StHMA*s. *StHMA*s are largely expressed in roots (*StHMA 1/2/5/18/24/27/35*) and stolons (*StHMA 6/13/14/17/21/26*), which may be connected with their possible role in heavy metal absorption and transport. Nevertheless, the transcription levels of these genes remain low in some organs. According to previous studies, the downregulation of some genes may be helpful in preserving their ancestral functions [52]. In other words, those *HMA*s may be conserved from their ancestors, which cannot be induced unless in certain special conditions appear.

Most HMA genes contain abiotic stress-related cis action elements [50,53]. When analysing the promoters of the *StHMA* family, we discovered 6 major promoter elements that function in response to abiotic stress (Figure 8). Transcription factors (TFs) and co-transcriptional regulators can precisely adjust the molecular response in various signaling cascades. Among them are various members belonging to the MYB, WRKY, NAC, DOF, AREB/ABF (ABA response-element binding factor), GBFs (G-box binding factors), and AP2/ERF families [54]. Therefore, this study investigated the expression profiles of 15 *StHMA* family members in different organs in response to various abiotic stresses. Only a few *StHMA*s react to all abiotic stresses, as *StHMA16*, *StHMA19,* and *StHMA36* actively respond to all six abiotic stresses (Figure 9). When stressed by Cd^2+^, 15 *StHMA* genes respond actively. When processed with PEG6000, an active response is detected only in *StHMA19* and *StHMA36*. Against flood stress, six *StHMA*s respond actively and are primarily expressed in the stem and root. When processed with Na^+^, 12 *StHMA*s present an active response, while *StHMA 16/19/31* are significantly expressed in different parts and are expressed almost 30-fold more strongly than that in the control group. For high-temperature stress, significant expression was observed in *StHMA2*, *StHMA19*, and *StHMA31*, and for low-temperature stress, 15 *StHMA*s were significantly expressed. Previous studies on the HMA gene family have focused on the mechanism of heavy metal transport. Our results show that a variety of abiotic stresses may induce the expression of StHMA family genes. Plants exposed to different abiotic stresses cause significant changes the cellular redox status (leading to an increase in reactive nitrogen), the HMA family may play an important role, as they are necessary for plants to survive in extremely oxidizing environments [55], and play a signaling role in mediating defense responses, including genes in the heavy metal associated (HMA) domain [55]. With respect to plant perception of different stresses inducing a coordinated response mediated by complex signaling networks, plants specifically sense environmental cues and consequently activate signaling cascades for assembling an overall response with the final aim of surviving [31]. We speculate that the HMA gene family is also closely related to abiotic stress. However, little information is available about their roles in the protection of plants against pathogens and environmental stresses. *AcHMA1* in the yeast expression system can improve yeast cell resistance to such stresses as salt, alkali, drought, and oxygen [56]. Previous studies have shown that some similar proteins play a role in the transportation of other heavy metals and together with antioxidative enzymes are responsible for cross tolerance mechanism. They participate in plant adaptation to other stresses [35,36,57]. The HIPP gene family containing HMA domain has some preliminary studies on abiotic stress. *HvFP1* from barley shows a complex expression pattern with induction under different abiotic stress conditions (e.g., cold, drought, and heavy-metal exposure) during leaf senescence and in response to abscisic acid [34]. Like *HvFP1*, *HIPP26* from Arabidopsis can also be induced by cold, salt, and drought stresses [58]. This study performed a comprehensive analysis of StHMA family genes against abiotic stress for the first time and provided a basis for further research investigating the functions of HMA genes faced with abiotic stress.

## 4. Materials and Methods

### 4.1. Plant Growth Conditions and Treatments

*S. tuberosum* cultivar “*YunShu505*” plants were micropropagated in vitro on Murashige and Skoog (MS) medium plus 30 gL^−1^ sucrose and 0.8% agar (Sigma-Aldrich, St. Louis, MO, USA), with the pH being adjusted to 5.8. Potato tissue culture seedlings were transplanted in the seedling tray after 3 weeks. Two months later, the plants with uniform growth and health were selected for processing (which was repeated three times). The plants were processed with 40 °C, 4 °C, 30% PEG6000 (mass ratio), 25% NaCl (mass ratio), and 1 g/L CdCl_2_. 500 mL processing liquids(drought stress, 30% PEG6000; salt stress, 25% NaCl; heavy metal stress, 1 g/L CdCl_2_) were uniformly sprayed on the plants with a watering pot (For simulating rain irrigation). High temperature stress (at 40 °C) and cold stress (at 4 °C) experiment were finished in artificial climate incubator (TAISITE RGX-400E, Tianjin Taisite Instrument Co., Ltd., Tianjin, China) The plants grown under a 16 h: 8 h, high light intensity: darkness regime. Samples were collected at 0 h, 6 h, 12 h, and 24 h after processing. In the control group, plants were sprayed with 500 mL clean water at 25 °C and sampled at corresponding time points. The 2^nd^ to 3^rd^ leaf, full stem, and root were cut from the *S. tuberosum* plants for quick freezing with liquid nitrogen and preservation at −80 °C for later use.

### 4.2. RNA Extraction and Quality Control

Total RNA was extracted from the *S. tuberosum* processed as mentioned above by following the manufacturer’s instructions for the RNAplant (RTR2303, Real-Times (Beijing) Biotechnology Co., Ltd., Beijing, China) reagent kit. RNA purity (OD_260_/OD_280_), concentration and integrity were tested with an ultralow volume nucleic acid tester NanoDrop 2000 (Shanghai Chuangmeng Biological Technology Co., Ltd., Shanghai, China) and an Agilent 2100 Bioanalyzer (Agilent Technologies, Palo Alto, CA, USA). The RNA passing the test was preserved at −80 °C for later use.

### 4.3. Identification, Physicochemical Properties Analysis, Chromosome Distribution Characteristics and Collinearity Analysis of the HMA Gene Family in S. tuberosum

*S. tuberosum* genome data files were downloaded from the plant genome EnsemblPlants database (http://plants.ensembl.org/Solanum_tuberosum/Info/Index) [59]. From the Pfam database (http://pfam.xfam.org/) [60], the hidden Markov Model (PF00403) of HMA structural domains was acquired and then combined with HMMBUILD and HMMSEARCH in localized HMM (http://www.hmmer.org/) to search candidate members for the HMA family from *S. tuberosum* with default thresholds. For HMA structural domains of candidate family members, SMART (http://smart.embl-heidelberg.de/) and NCBI CDD (https://www.ncbi.nlm.nih.gov/cdd/) were applied for further integrated analysis and identification to determine the final members of the HMA gene family. The screened HMA gene family members of *S. tuberosum* were analysed in terms of physicochemical properties, such as amino acid number in sequence, gene position, theoretical isoelectric point, and molecular weight, using ExPASy (http://expasy.org/) [61]. Online software MG2C (http://mg2c.iask.in/mg2c_v2.0/) was employed to plot the chromosome distribution of the HMA gene family. Genome-wide sequences of *A. thaliana*, *I. triloba*, *N. attenuata*, *Z. mays* and *O. sativa* were downloaded from EnsemblPlants database (http://plants.ensembl.org). The detection and study of the gene duplication events in HMA genes were per formed with the use of multiple collinear scanning toolkits (MCScanX) [62]. Corresponding results were visualized using Circos software [63]. A computation of their Ka (Synonymous), Ks (Nonsynchronous substitution), and their ratio with KaKs_Calculator2.0 software [64].

### 4.4. Multi-Sequence Alignment of HMA Gene Family and Phylogenetic Tree Construction in S. tuberosum

To determine the evolutionary relations among members of the HMA family in *S. tuberosum*, amino acid sequences of *A. thaliana*, *O. sativa* and *S. lycopersicum* were acquired from The Ensembl Genome Browser [59] and aligned with that of *S. tuberosum* in HMA amino acid sequences. The alignment results were used to form a phylogenetic tree using Neighbor-Joining [65] 1000 bootstrap in MEGA X software [66].

### 4.5. Genetic Structure and Motif Analysis of the HMA Gene Family in S. tuberosum

After CDS region sequences and HMA genome-wide sequences of StHMA family members were retrieved from the phytozome database, online software GSDS2.0 (http://gsds.cbi.pku.edu.cn/index.php) [67] was used to predict the distribution of their introns, CDS region, and 5’ and 3’ untranslated regions. Conversely, the motif structures of HMA family members in S. tuberosum were forecast with MEME (http://meme-suite.org/) with the number of target motifs set to 15 [68,69]. Their intron and exon sequences were analysed and visualized with TBtools.

### 4.6. Cis-Element and Expression Analysis of StHMA Gene Family

The expression profiles of 36 members in the *StHMA* family were acquired from the gene expression profile database (ArrayExpress, https://www.ebi.ac.uk/arrayexpress) (accession number: E-MTAB-552) [70]. In this study, an analysis was performed on the expression levels of 36 genes in different tissues or organs against diverse stresses. Heatmaps were generated with HemI from the normalized value by row for the signatures in transcripts per million (TPM). The analytic results were visualized using TBtools software in the form of a heat map [71].

From *S. tuberosum* genome data files (http://plants.ensembl.org/Solanum_tuberosum/Info/Index) [59], the upstream 2000 bp sequences of transcription start sites of all the members were downloaded for a predictive analysis of abiotic stress-related cis acting elements with TBtools (V.1.046) [71]. Many studies have indicated massive functional redundancy among StHMA members. Therefore, based on the phylogenetic tree, 2–3 members were chosen from the *StHMA* family to undergo an expression analysis with real-time quantitative PCR against abiotic stress. With CDS sequences of the HMA gene family acquired from the *S. tuberosum* genome database, primers (Appendix A) were designed with Primer6.0 software and then delivered to SANGON Biotech (Shanghai, China) for synthesis. The template cDNA synthesis was finished with Primer ScriptTM RT regent Kit and gDNA Eeaser kit (TaKaRa, Beijing, China) during the quantification. With root, stem and leaf sampled at 0 h, 6 h, 12 h and 24 h, respectively, after processing, as the raw materials, total RNA was extracted and reverse transcribed into cDNA. With the *S. tuberosum EF1α* (elongation factor *1-alpha*) gene as the internal reference, the relative expression of each gene in all parts of *S. tuberosum* was determined using the 2^−ΔΔCt^ method.

## 5. Conclusions

HMA genes are involved in many biological processes in plants. In this paper, we identified 36 StHMAs from *S. tuberosum*. Gene replication analysis indicated that HMA genes tend to be conserved in different plants, and some of them may come from the same ancestor. Based on phylogenetic analysis, the HMAs were grouped into six subfamilies. HMA genetic structures and conserved domains vary greatly with species. As revealed by the expression profile analysis, *StHMA* expression is tissue-specific and plays a role in Cd stress, salt stress, and some abiotic stresses. This study may help to elucidate the biological functions of *StHMA*s.

## Figures and Tables

**Figure 1 genes-11-01269-f001:**
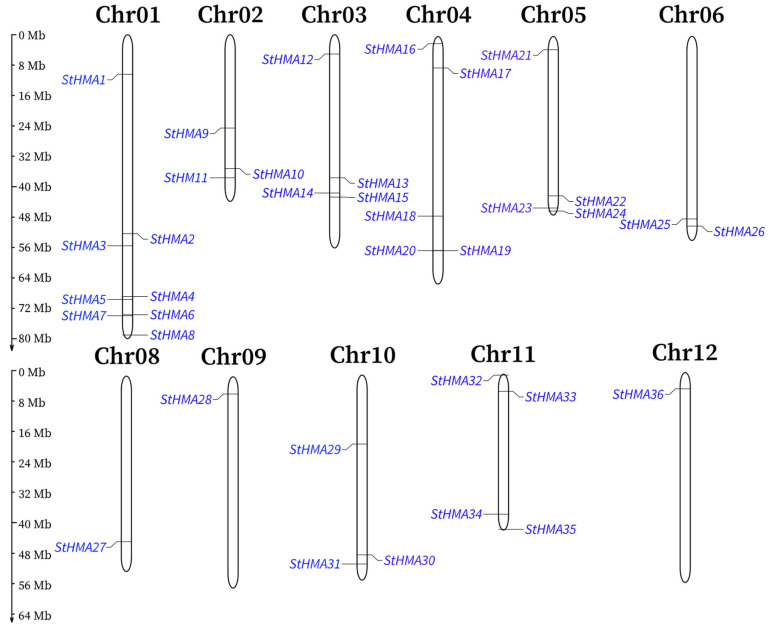
Chromosomal locations of *StHMA* genes in *Solanum tuberosum*. Black bars represent the chromosomes. Chromosome numbers are shown at the tops of the bar. *StHMA* genes are labeled at the left and right of the chromosomes. Scale bar on the left indicates the chromosome lengths (Mb).

**Figure 2 genes-11-01269-f002:**
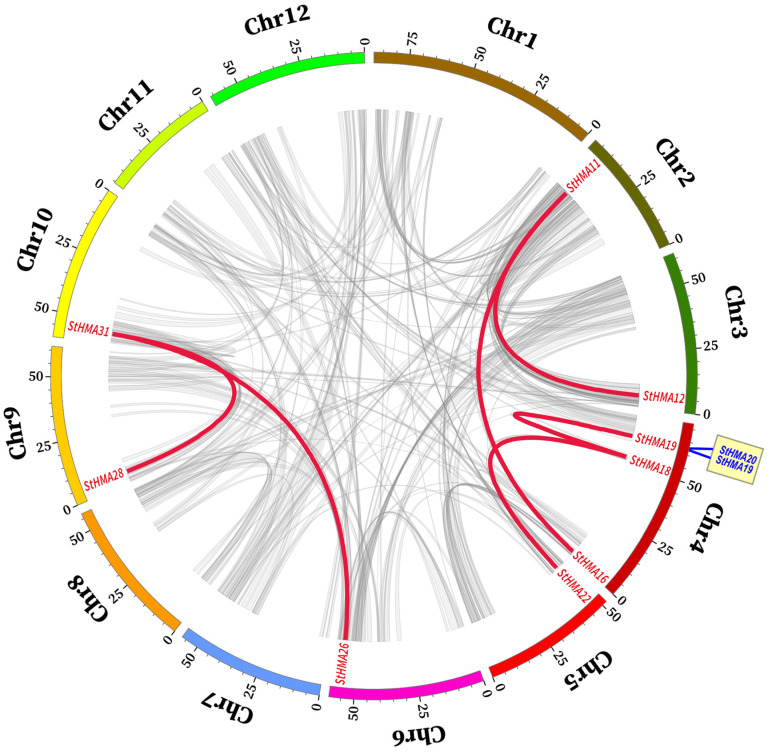
Schematic representations of segmental duplications of *Solanum tuberosum* StHMA genes. Different gray lines indicate all synteny blocks in potato genome between each chromosome. Thick red lines indicate segmental duplication heavy-metal-associated gene pair, and thick blue lines indicate tandem duplication gene pair. The chromosome number is indicated at the bottom of each chromosome. Scale bar marked on the chromosome indicating chromosome lengths (Mb).

**Figure 3 genes-11-01269-f003:**
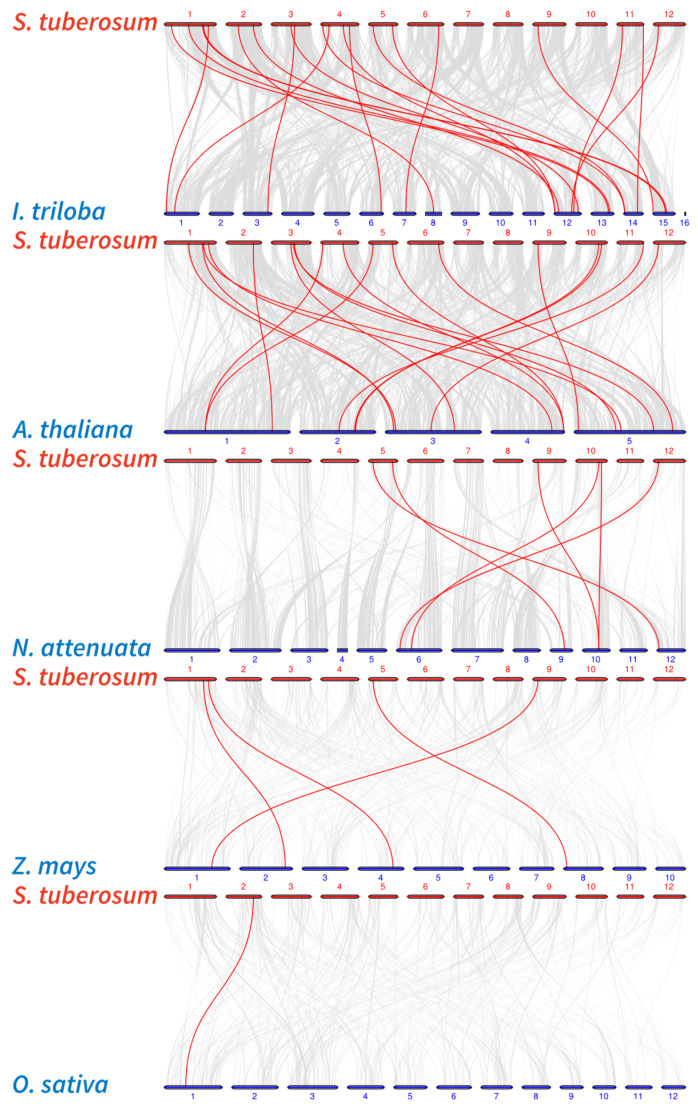
Collinearity analysis of heavy-metal-associated genes between *Solanum tuberosum*, *Ipomoea triloba*, *Arabidopsis thaliana*, *Nicotiana attenuata*, *Zea mays*, and *Oryza sativa*. Gray lines in the background indicate the collinear blocks within potato and other plant genomes, while the red lines highlight the syntenic *StHMA* pairs of orthologous genes. The species names with the prefixes “*I. triloba*”, “*A. thaliana*”, “*N. attenuate*”, “*Z. mays*”, and “*O. sativa*” indicate *Ipomoea triloba*, *Arabidopsis thaliana*, *Nicotiana attenuata*, *Zea mays*, and *Oryza sativa*, respectively. Blue color bars represent the chromosomes of different species. The chromosome number is labeled at the top or bottom of each chromosome.

**Figure 4 genes-11-01269-f004:**
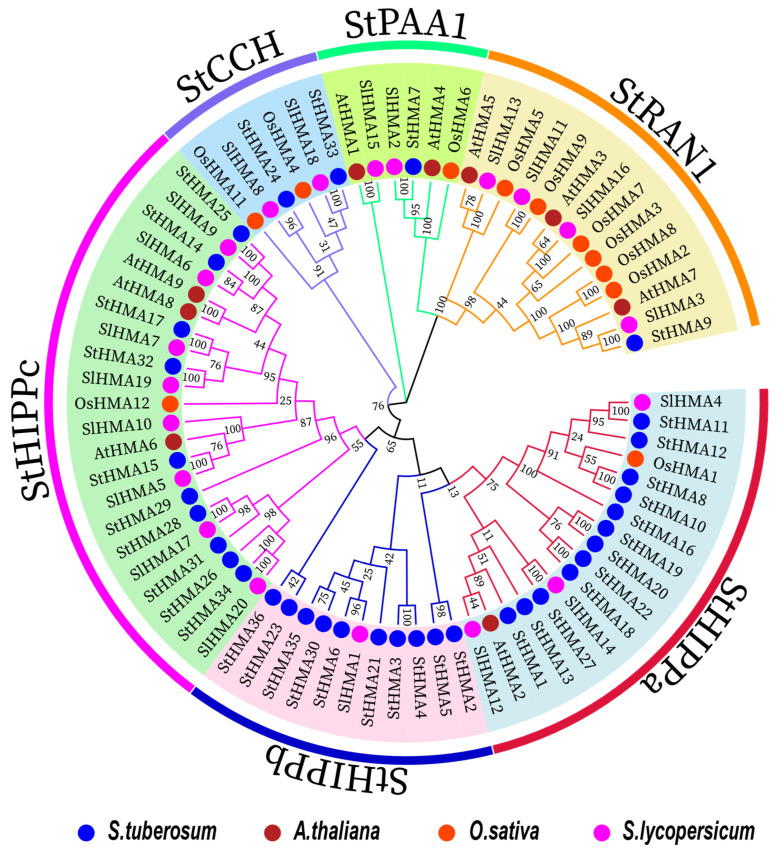
Phylogenetic relationships among 78 heavy-metal-associated proteins in *Solanum tuberosum*, *Arabidopsis thaliana*, *Oryza sativa* and *Solanum lycopersicum*. The maximum likelihood tree was created using MEGA X (bootstrap value = 1000) and the bootstrap value of each branch is displayed. The blue solid circles, brown solid circles, red solid circles and pink solid circles represent heavy-metal-associated proteins from potato, *Arabidopsis*, rice and tomato, respectively.

**Figure 5 genes-11-01269-f005:**
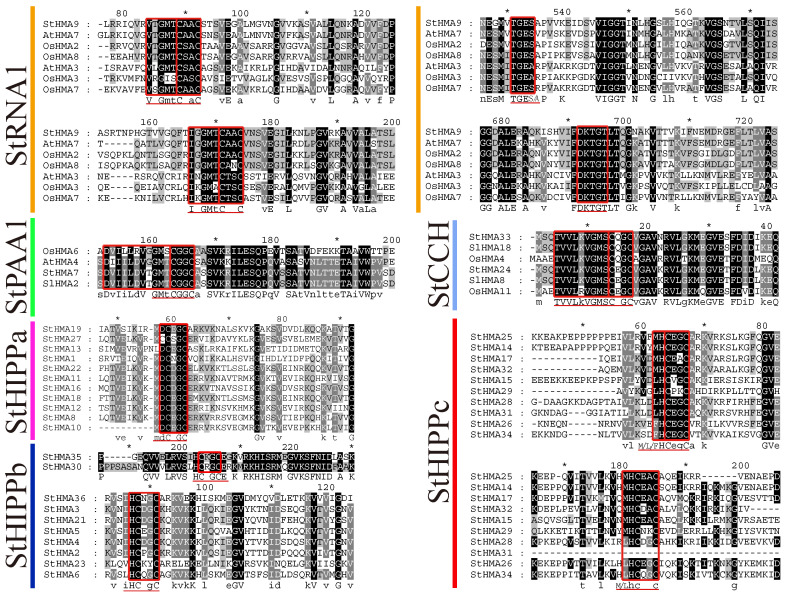
Multiple Alignment of StHMAs Metal binding domains amino acid sequences. “StRNA1”, “StPAA1”, “StCCH”, “StHIPPa”, “StHIPPb”, and “StHIPPc” represent different StHMA proteins classification. The putative xxCxxC, TGEx, and DKTGT motifs locations were highlighted in red boxes. Comparison of MBD motifs in *Solanum tuberosum*, *Arabidopsis thaliana*, *Oryza sativa* and *Solanum lycopersicum* HMA proteins.

**Figure 6 genes-11-01269-f006:**
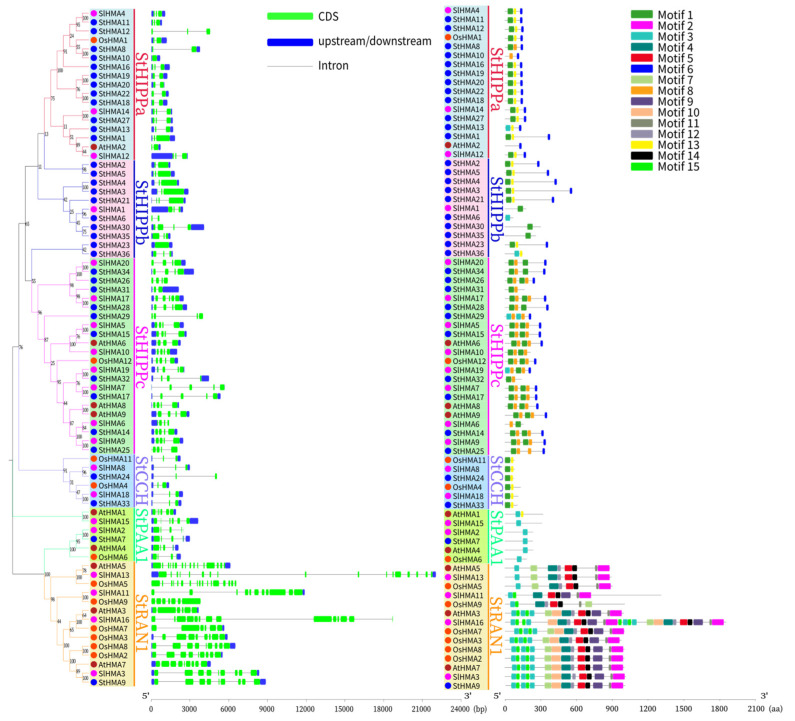
Exon-intron structure, phylogenetic relationships, and protein domain analysis of StHMAs genes in potato. Phylogenetic relationships and exon-intron structure of StHMA genes. Green lines indicate introns. Blue boxes represent upstream/downstream-untranslated regions. The scale bar at the bottom estimates the lengths of the exons, introns, and untranslated regions. Phylogenetic relationships and motif structures of HMA proteins between *Solanum tuberosum*, *Arabidopsis thaliana*, *Oryza sativa* and *Solanum lycopersicum*. The maximum likelihood tree was generated with MEGA X using the NJ method. StRAN1, StPAA1, StCCH, StHIPPa, StHIPPb, and StHIPPc are marked with different colors. Motif analysis was performed using the MEME program. Boxes of different colors represent the various motifs. Their location in each sequence is marked. Motif sequence logo is shown in Appendix A. The scale bar at the bottom indicates the lengths of the HMA protein sequences.

**Figure 7 genes-11-01269-f007:**
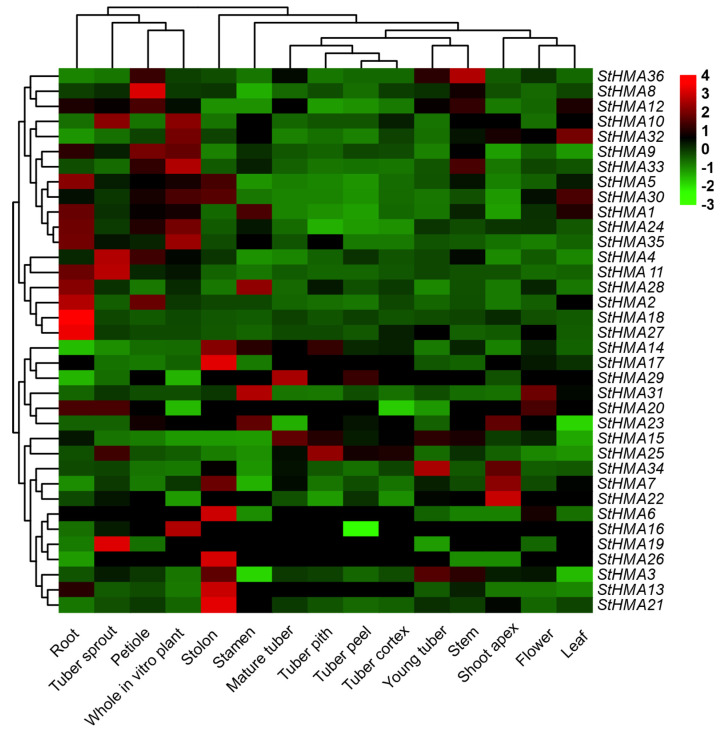
Tissue-specific expression profiles of StHMA genes in different *Solanum tuberosum* organs. The transcription levels of StHMA genes in different tissues. Heatmaps were generated using the TBtools (Toolbox for biologists V.1.046) software from the normalized value by row for the signatures in transcripts per million (TPM), and gradient color from green to red was transcript levels of each StHMA gene. StHMA family were acquired from the gene expression profiles database (ArrayExpress, https://www.ebi.ac.uk/arrayexpress) (accession number: E-MTAB-552), are listed in Appendix A.

**Figure 8 genes-11-01269-f008:**
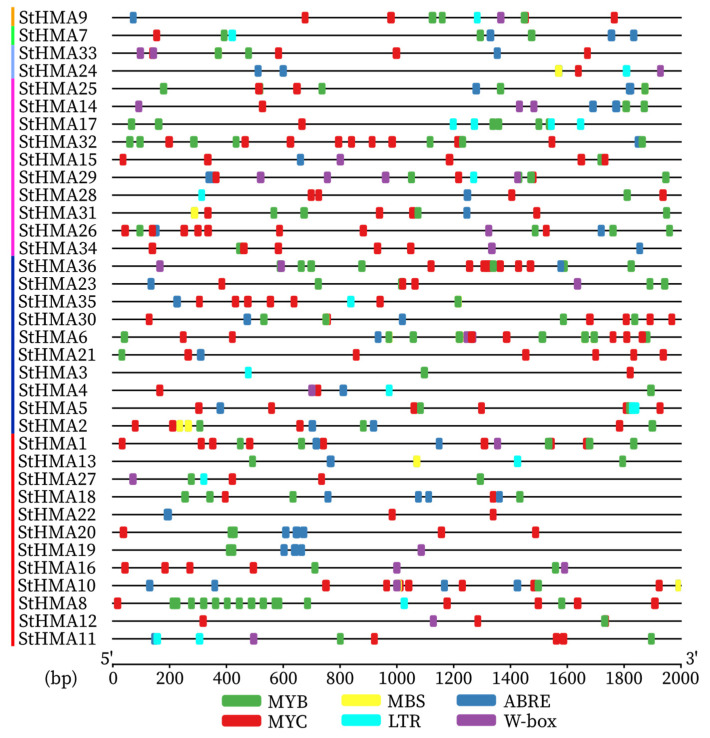
Predicted cis-elements in the promoter regions of the *Solanum tuberosum* StHMA genes. Abiotic stress-related promoter sequences (−2000 bp upstream genomic sequence) were analyzed. The scale bar at the bottom indicates the length of promoter sequence. Different cis-elements were labeled by rectangle of different color. Abiotic stress-related cis-acting elements are shown in Appendix A.

**Figure 9 genes-11-01269-f009:**
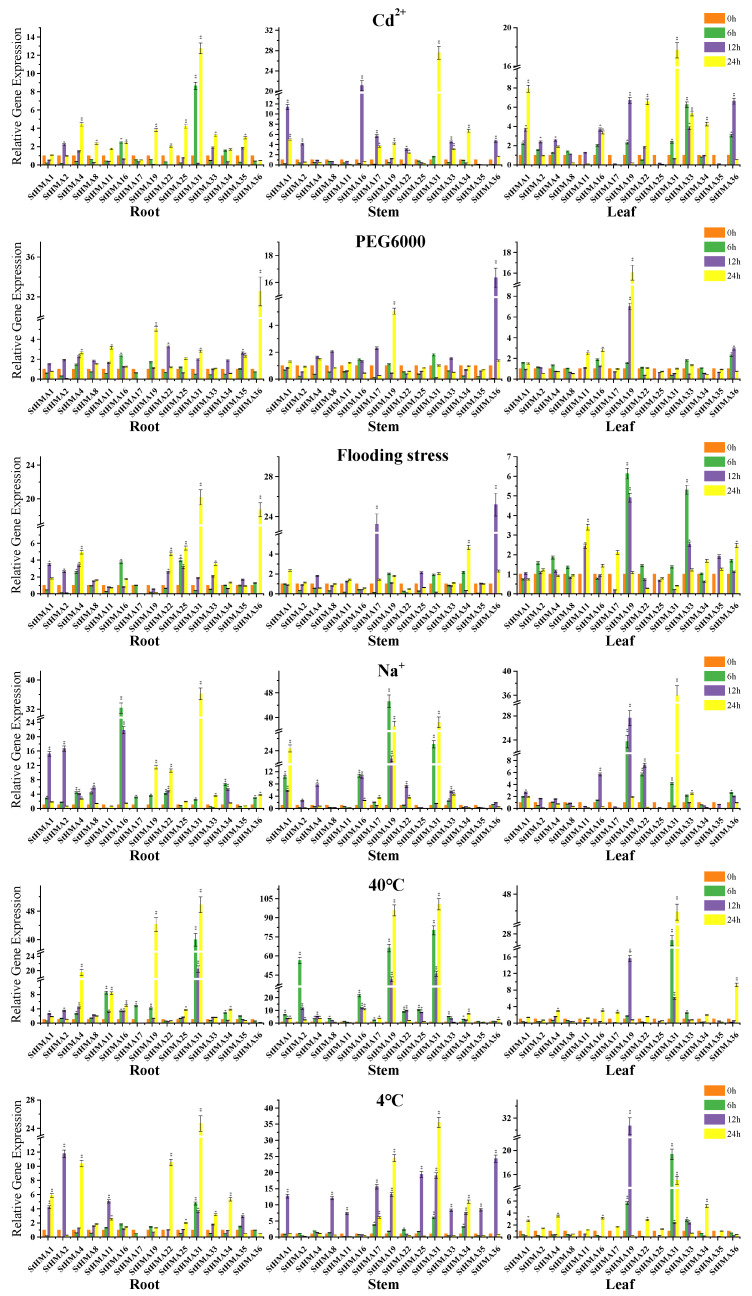
Expression profiles of 15 selected *StHMA*s in response to 40 °C, 4 °C, 30% PEG6000 (mass ratio), 25% NaCl (mass ratio), and 1g/L CdCl_2_ stresses in 2-months old potato seedlings after treatment for 6, 12, and 24 h. Data represent means (±SD) of three biological replicates. Vertical bars indicate standard deviations. Asterisks indicate corresponding genes significantly upregulated or downregulated between the treatment and control (n = 15, * *p* < 0.05; ** *p* < 0.01; Student’s *t*-test).

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
