# Peer review of "Heavy Metal Transporters-Associated Proteins in Solanum tuberosum: Genome-Wide Identification, Comprehensive Gene Feature, Evolution and Expression Analysis"

_genes, 2020, doi:10.3390/genes11111269_

Round 1
Reviewer 1 Report
Comments to MS „Heavy metal transporters-associated proteins in Solanum tuberosum: genome-wide……… ” written by Guandi He et al.
It was demonstrated many times that heavy metal stress and resulting level of reactive oxygen species play an important regulatory role in aclimatory and defence mechanisms in all plants, however, our knowledge about response of regulation of dedicated gene in different plant tissues during application of Cd is rather scant. The general intention of this MS is good and the problem is well stated. Results presented in this MS have big scientific potential and there are several novel aspects of this work that extend beyond what has previously been published in the literature. This manuscript focuses on HMAs proteins participating in heavy metal detoxification. We know also that these genes play important role in plants exposed to other stresses. There are some issues the authors should address before this manuscript could be considered acceptable. All stated details dealing with the structure and evolution of these proteins need interpretation and first of all they need adequate physiological interpretation.
Key words: It is not necessary to repeat expressions used in the title.
Abstract: Some physiological statements resulting from all analyses should be added.
Solanum tuberosum should be written in “italic”.
Introduction:
Potential reader of this MS will expect the meaning of all stated facts on plant resistance to Cd and other stresses. Recently (2019-2020) several papers were published (Plants and Journal of Plant Physiology) describing Cd-caused effects and among other HMAs protein expression. Expression of HMAs depend on physiological plants status and plant tissue. How do you know if stress applied to leaves can change HMA expression in roots? Despite of this I would advise to describe with more details exposure of your plant material to different stresses (amount of warm and cold water used for processing of plants).
Major criticism:
It is well known that changes in activity of metal transporting proteins are crucial for plant resistance to cadmium. Their role in plant reaction to other stresses need to be elucidated. In this MS we learn a lot about the location and structure of these proteins. We know that some similar proteins play a role in transportation of other heavy metals and together with antioxidative enzymes are responsible for cross tolerance mechanism. They participate in plant adaptation to other stresses. This would help to understand plant reaction in given circumstances, and also why plants in this MS were exposed also to other stresses.
Discussion should also point out the novel aspects of presented results for physiology of plants. Authors of this paper should not present possible increase in expression of these proteins only in terms of stress.
Reviewer 2 Report
This study presents a comprehensive investigation of HMA family genes in S. tuberosum. The topic is interesting, the analysis is solid and the results will help better understand this gene family in S. tuberosum. I only have some minor suggestions
Line 79, “With S. tuberosum cultivar “YunShu505” plants, as the research subject, this study sets out to analyze the genetic structure, chromosome distribution and phylogenetic relationship of HMA family …”
I found this sentence is misleading, as the authors actually did the above analysis using the reference genome, not “YunShu505”
Line 86, PF00403 looks like a PF domain, is it the HMA domain ? why this information was provide after hidden Markov Model? What is HMA domain ?
In line 128, “and 20 and 18 paralogous genes
129 with 9 chromosomes of I. triloba and 5 chromosomes of A. thaliana, respectively” is not clear.
In line 132, “paralogous genes” by definition are genes from the same species, authors should learn the difference between paralogous and Orthologous
In line 133-135, it is a brave speculation, I didn’t see enough evidence support the so-called massive gene multiplication
In line 187, not sure what the author want to say with “However, we make no distinction in this study”
Round 2
Reviewer 1 Report
Comments to MS “Transcriptome analysis reveals the regulation of jasmonic acid and carbon/nitrogen assimilation in drought tolerance of rapeseed (Brassica napus L.)” written by Jihong HU et al. .
In literature there are a lot of publications describing changes in expression of some proteins their activity and changes in expression of related genes in plants exposed to different abiotic stresses. This was checked in roots and leaves. This manuscript focuses on adaptive changes in plants exposed to drought. The role of JA in adaptation to stress was described in details 30-40 years ago. Changes in antioxidative system and proteins responsible for carbon and nitrogen fixation in plants exposed to different environmental factors and their adaptation to fluctuating environment was described in many other papers, however, our knowledge about the way of their regulations just in chosen B.napus lines is rather scant. Authors of the paper have done experiments with this interesting plant material. Thus, the analysis of the effects caused by stress factors could be of practical meaning for applied sciences. There are some novel aspects of this work that extend beyond what has previously been published in the literaturÄ™ and this could be exposed much more clearly basing on obtained here results.
However there are some issues the Authors should address before this manuscript could be considered acceptable. To my opinion this MS would be more interesting for potential reader of GENES as short communication, as physiological questions are, in fact, not discussed and correlation of expression of different genes is similar to observed in other plants Sequence of events after exposure to drought would be new and interesting for potential reader. It is also difficult to draw conclusions from presented data. We can only learn that these genes are involved in stress reaction as in other plants. While preparing this MS as short communication some data should be presented as supplementary data.
Major criticism:
Title:
- should be not in “italic”.
Abstract: I would characterize shortly chosen plants as this is in fact new in this MS.
Key words: It is not necessary to use expressions used in the title.
Introduction: Carbon and nitrogen assimilation is usually going parallel with reaction to stress. You should underline what is not very typical in this plant material.
L.53 English
M&M
The description of material and methods is sufficient judge and repeat the experiments.
However, exposure to drought, what is essential in this work, should be described with more details and more variants of exposure to drought should be involved in these experiments to be sure that observed effects result just from this kind of stress.
Discussion: In fact, it is not correct to interpret all changes as indicative for stress. Some changes can be discussed in terms of adaptation or signaling process.
Despite the above criticism I would say that this paper after correction can be very nice contribution to Genes. Generally, the paper is concisely written.
Author Response
The original ID:genes-961899-comments are not corresponding to the manuscript.
We are sorry to bother you. We found the comments are for paper with a title of "Transcriptome analysis reveals the regulation of jasmonic acid and carbon/nitrogen assimilation in drought tolerance of rapeseed (Brassica napus L.)", but our title is "Heavy Metal Transporters-Associated Proteins in S. tuberosum:Genome-wide Identification, Comprehensive Gene Feature, Evolution and Expression Analysis". Please kindly check again.
